# FEM Analysis of 3D Timber Connections Subjected to Fire: The Effect of Using Different Densities of Wood Combined with Steel

Elza M. M. Fonseca * and Carlos Gomes

ISEP, Instituto Politécnico do Porto, R. António Bernardino de Almeida, 4249-015 Porto, Portugal;
1170942@isep.ipp.pt
* Correspondence: elz@isep.ipp.pt

**Abstract:** This work aims to present a study approach for double-shear connections of wood under fire with dowel pins and plates in steel material, using different types of glulam. The simplified Eurocode equations for ambient temperature were used to determine the dimensions and the number of dowel pins that each studied connection needs in order to resist an applied tensile load. Following this methodology, the finite element method was used to assess the thermal analysis of the studied connections under fire. The study aims to increase the information on these connections, where the wood material represents a complicated behavior in fire circumstances, with the addition of the steel material. The heat conducted by the dowel pin inside the connection, and the steel plate and its effect on the wood were analyzed. According to the results, it can be assumed that the temperature evolution is due to the geometry of the connection, the dowel pin or plate position, and the glulam density. Inside the wood element, the temperature remains lower, and externally a charred depth is developed when the target temperature of 300 °C is reached, and, in the vicinity of the dowel pin or the steel plate, a burned wood depth is indirectly formed. The rate of the charred layer is not constant throughout the entire fire exposure. Steel-to-timber connections with an internal steel plate with high glulam density have greater fire resistance due to the lower temperatures obtained.

**Keywords:** steel-to-timber connection; timber-to-timber connection; fire

## 1. Introduction

With the growing concern about climate change, the use of wood in buildings presents itself as a solution to this problem. The main justification is related to the positive contribution to the carbon cycle, and, in addition to this aspect, wood can present good fire resistance characteristics as well as mechanical strength and density [1,2].

Wood is a sustainable material and can contribute to combating the current problem of climate change, but to the detriment of the use of other materials [1,2].

In addition to these characteristics, the wood has good fire resistance and a wide range of applications in construction. For these reasons, the interest in this subject within the scientific community has increased, making it very useful for assessing the structural safety of these elements and in updating current standards, which are governed by easy equations that can lead to the over-dimensioning of elements. In addition, in terms of fire protection, the standards require a high level of safety, so wooden structures must be designed with precision.

Connections are considered the critical elements of wood construction, transmitting high loads, and demanding greater importance during the design steps. However, when adding fire exposure, connections can be the vulnerable points of the construction. After this literature review, it was concluded that there are still very few studies on fire resistance in wooden elements compared to studies on the use of other materials, such as steel or concrete.

Different studies involving analytical and experimental models have been developed to calculate temperature results on the performance of connections exposed to fire conditions [1,2]. More recently, studies involving numerical models have been developed to predict fire exposure in connections [3–6].

To increase the fire resistance time for timber connections, Eurocode 5 part 1–1 [7] proposes, for timber connections, increasing the thickness and width of the lateral members, as well as the distance to the edges of the dowels. In lateral elements directly exposed to fire, there is a need for the protection of the members. In this work, the objective is to study connections without any material for fire protection.

The core of this study is to present an analytical and numerical formulation that can be applied in the project of glulam double-shear connections in conjunction with steel material, joined by dowel pins, as represented in Figure 1. The selection of the specific dimensions depends on the type of timber connection and the necessary load-carrying capacity. Fastener strength depends on many factors, such as externally applied load, wood density, load direction of the wood grain, position of the fasteners, edge, and distances from the ends of the timber connection [6].

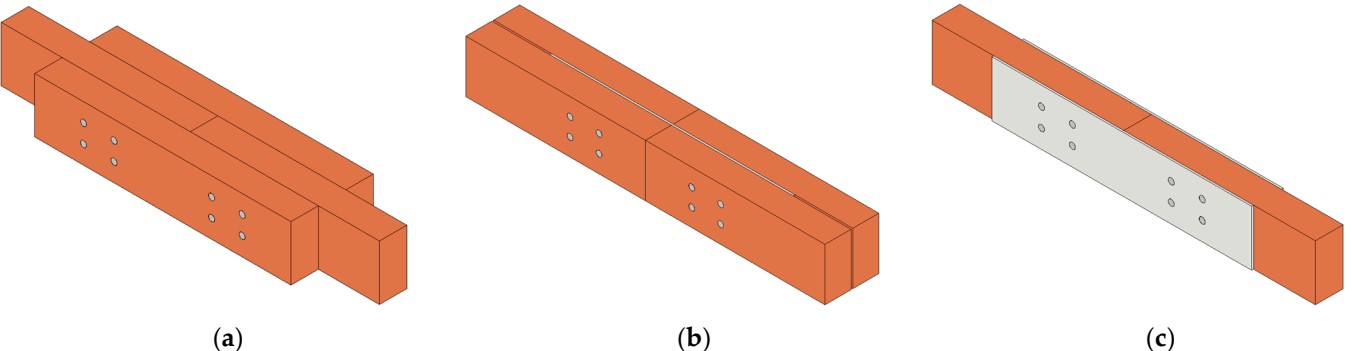

| (a) | (b) | (c) |

**Figure 1.** Connections: (**a**) timber-to-timber, (**b**) steel-to-timber with internal steel plate, (**c**) steel-to-timber with external steel plate.

The presented analytical procedure follows the simple equations of Eurocode 5 part 1–1 [7] to determine the dimensions of the connections. The numerical analysis uses the finite element method in the thermal and transient algorithm to obtain the fire behavior in each connection. The variation in the effect of the glulam density was also examined to characterize the thermal behavior and the evolution of the charring rate in the wood elements.

Following the previous works already developed by the authors [4,5], the present study gives the results in complete 3D timber connections simulated with thermal solid finite elements, instead of 2D finite elements in which only the 1D heat effect is observed in the cross-sectional plane.

The 3D models thus appear as close as possible to real connections, where the entire set of elements collaborates in the different thermal functions.

With this 3D study, it was possible to visualize and calculate the heating inside the connections, and, at the same time, determine the temperatures developed in the dowel pins in addition to the charred area in the wooden elements.

## 2. Materials and Methods

### 2.1. Mechanical Properties

Table 1 represents the mechanical properties of the three different glulam densities used for the designed connections, and the S275 steel material for the other connectors (dowels pin and plates), according to the references [4,8–10].

**Table 1.** Mechanical properties of the glulam and steel.

| Mechanical Properties | GL20H | GL24H | GL32H | S275 |
|---|---|---|---|---|
| Yield strength, MPa | 16 | 19.2 | 25.6 | 275 |
| Young's Modulus, GPa | 8.4 | 11.5 | 14.2 | 210 |
| Poisson ratio | 0.4 | 0.4 | 0.4 | 0.3 |
| Density, kg/m$^3$ | 370 | 420 | 480 | 7850 |

*2.2. The Load-Carrying Capacity in the Connection*

Eurocode 5 part 1–1 [7] provides easy equations to calculate the load-carrying capacity in various connection types. In these connections, the basic requirement for the calculation is the symmetrical arrangement between all elements [4,5,11]. For the design of the connection, the aim is to determine the number of dowels and their dimensions.

For this work, different parameters were considered: one external tensile load $F_d$ equal to 10 kN, the dowel diameter $d$ equal to 10 mm, the wooden board thickness $t_1$ and $t_2$ equal to 45 mm, and the steel plate thickness $t_s$ equal to 3 mm. The following equations are used for double-shear connections, where the dowel diameter must be greater than 6 mm and less than 30 mm [4,5]. We take into consideration that $A_s$ is the cross-section of the member and the design tensile stress along the grain is $\sigma_{t,0,d}$.

$$\sigma_{t,0,d} = \frac{F_d}{A_s} \tag{1}$$

The characteristic load-carrying capacity per shear plane and fastener $F_{v,Rk}$, is obtained according to Eurocode 5 part 1–1 [7]. To include effects dependent on connection geometry, wood embedment strength, and fastener bending, the characteristic load-carrying capacity will be the smallest value from the Equations (2) to (4) [4,5,7,11–13].

For a steel-to-timber connection with dowel pins, $F_{v,Rk}$ is given by the equations in (2), when the steel is central to the wooden members.

$$F_{v,Rk} = min \begin{cases} f_{h,1,k}t_1d & (a) \\ f_{h,1,k}t_1d\left[\sqrt{2 + \frac{4M_{y,Rk}}{f_{h,Rk}dt_1^2}} - 1\right] + \frac{F_{\alpha x,Rk}}{4} & (b) \\ 2,3\sqrt{M_{y,Rk}f_{h,1,k}d} + \frac{F_{\alpha x,Rk}}{4} & (c) \end{cases} \tag{2}$$

For a timber-to-timber connection with a dowels pin, the equations in (3) allow for the calculation of $F_{v,Rk}$.

$$F_{v,Rk} = min \begin{cases} f_{h,1,k}t_1d & (a) \\ 0.5f_{h,2,k}t_2d & (b) \\ 1.05\frac{f_{h,1,k}t_1d}{2+\beta}\left[\sqrt{2\beta(1+\beta) + \frac{4\beta(2+\beta)M_{y,Rk}}{f_{h,Rk}dt_1^2}} - \beta\right] + \frac{F_{\alpha x,Rk}}{4} & (c) \\ 1.15\sqrt{\frac{2\beta}{1+\beta}}\sqrt{2M_{y,Rk}f_{h,1,k}d} + \frac{F_{\alpha x,Rk}}{4} & (d) \end{cases} \tag{3}$$

For a steel-to-timber connection $F_{v,Rk}$ depends on the thickness of the external steel plates [5,7]. In the studied connections, the external steel plate is classified as a thin plate and is obtained using the equations in (4).

$$F_{v,Rk} = min \begin{cases} 0.5f_{h,2,k}t_2d & (a) \\ 1.15\sqrt{2M_{y,Rk}f_{h,2,k}d} + \frac{F_{ax,Rk}}{4} & (b) \end{cases} \tag{4}$$

In the equations, $f_{h,i,k}$ ($f_{h,1,k}$ or $f_{h,2,k}$) is the characteristic embedment strength in timber member $i$; $M_{y,Rk}$ is the characteristic yield moment of the fastener; $\beta$ is the ratio between

the embedment strength of the members considered equal to 1; and $F_{ax,Rk}$ represents the characteristic axial withdrawal capacity of the fastener [5,7].

The value of the characteristic embedment strength in the timber member $i$ parallel to the grain is defined in Equation (5).

$$f_{h,1,k} = f_{h,2,k} = 0.082(1 - 0.01d)\rho_k \tag{5}$$

The value of $M_{y,Rk}$ is determined corresponding to the dowel pin diameter $d$, and its material characteristic tensile strength $f_{u,k}$.

$$M_{y,Rk} = 0,3 f_{u,k} d^{2,6} \tag{6}$$

After calculating the value of $F_{v,Rk}$, we need to establish the design value of the characteristic load-carrying capacity, shown in Equation (7).

$$F_{v,Rd} = \frac{F_{v,Rk} k_{mod}}{\gamma_M} \tag{7}$$

The modification factor for load duration and moisture content $k_{mod}$ is equal to 0.6 for permanent load action, and the partial factor for material property $\gamma_M$ is equal to 1.25 for glulam [5,7].

In the connections studied, the values obtained for the characteristic load-carrying capacity are shown in Table 2.

**Table 2.** The characteristic load-carrying capacity in all connections $F_{v,Rd}$, kN.

| Connection Type | GL20H | GL24H | GL32H |
|---|---|---|---|
| Timber-to-timber | 2.57 | 2.85 | 3.19 |
| Steel-to-timber (internal steel plate) | 3.14 | 3.47 | 3.87 |
| Steel-to-timber (external steel plate) | 3.57 | 3.80 | 4.06 |

As calculated, a steel-to-timber connection with an external steel plate with high density material has a higher load-carrying capacity, and timber-to-timber with lower density material has a lower strength capacity. The characteristic load-carrying capacity increases with higher wood density.

Finally, with the calculation of $F_{v,Rd}$, the number of dowels is obtained with the following relationship, shown by Equation (8).

$$N \geq \frac{F_d}{F_{v,Rd}} \tag{8}$$

The layout between dowel pins corresponds to the calculated spacing, resulting from the equations, according to Eurocode 5 part 1–1 [7].

The number of calculated dowels in all connections studied was less than 4, and this number was considered to distribute across 2 rows and 2 columns. The calculated dimensions and positions are shown in Figure 2.

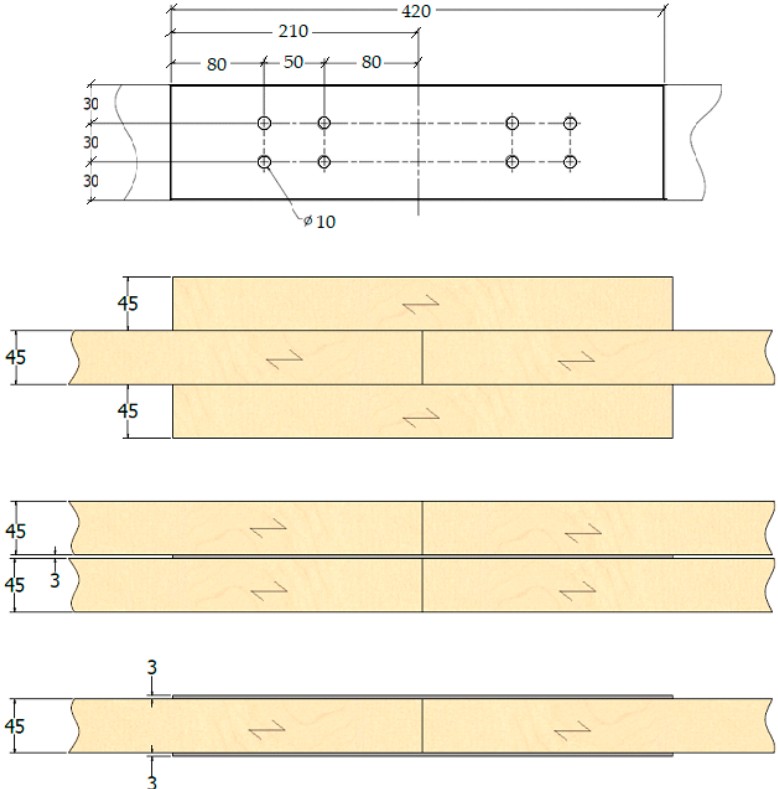

**Figure 2.** Connections' dimensions.

### 2.3. Thermal Properties

The thermal properties of wood and steel were considered (thermal conductivity, specific heat, and density) according to the Eurocode 5 part 1–2 [14], and according to the Eurocode 3 part 1–2 [15], respectively.

The steel emissivity is equal to 0.7 [15]. The emissivity of wood was taken to be equal to 0.8 [14]. Figures 3–7 present the thermal properties of wood and steel used in the numerical model, dependent on the temperature [4].

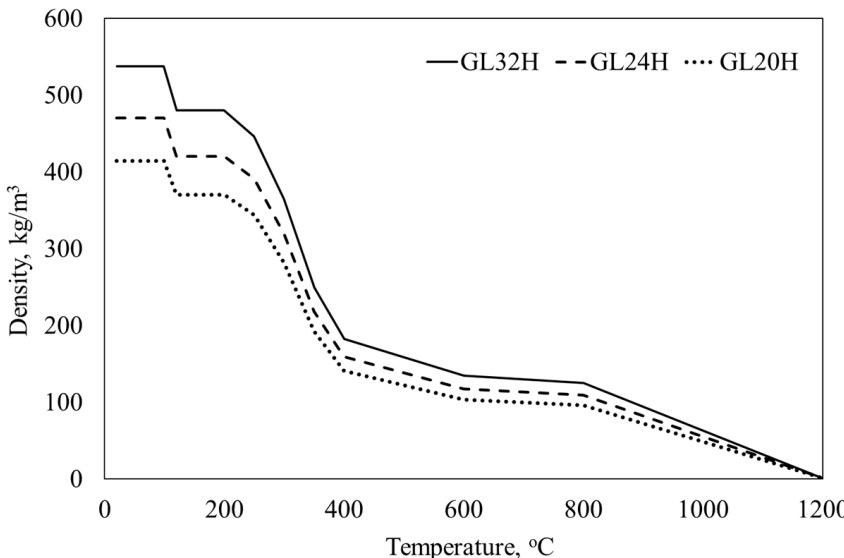

**Figure 3.** Variation in the density of the three different glulam types with temperature.

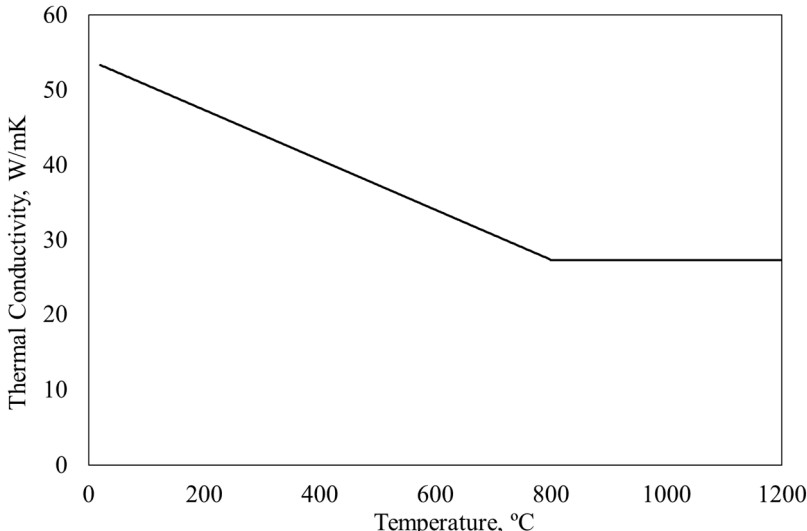

**Figure 4.** Variation in the thermal conductivity of steel with temperature.

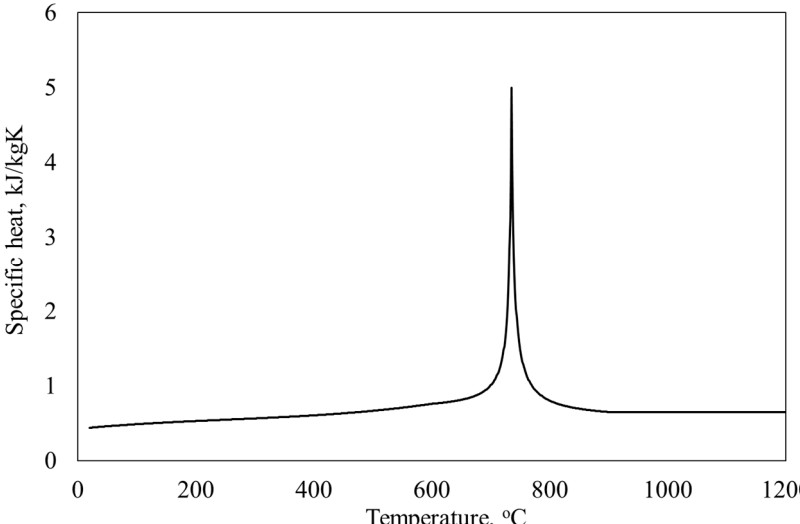

**Figure 5.** Variation in the specific heat of steel with temperature.

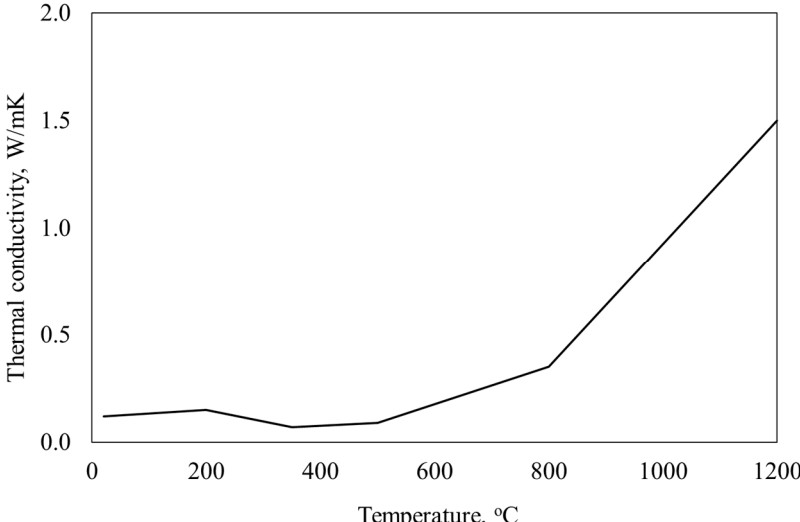

**Figure 6.** Variation in the thermal conductivity of wood with temperature.

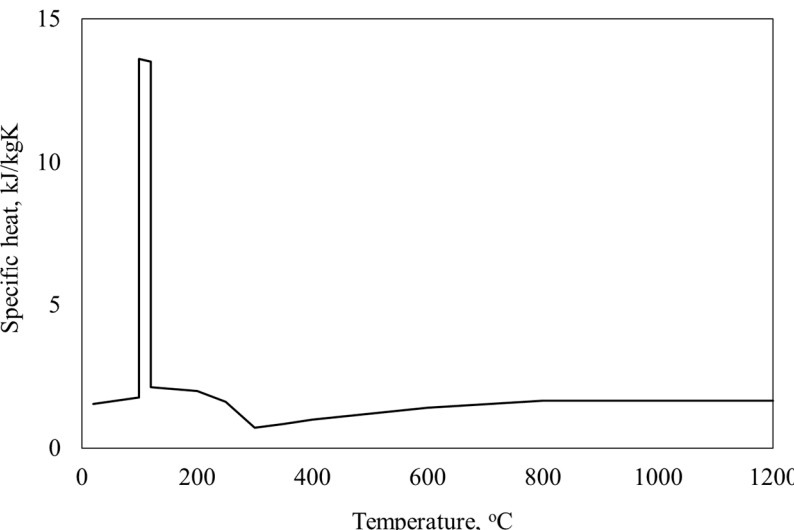

**Figure 7.** Variation in the specific heat of wood with temperature.

## 3. Thermal Model

The ANSYS® [16] program was used as a finite element method for transient and non-linear material thermal analysis.

The number of elements and nodes can influence the solution obtained and the decision of the number of finite elements through a convergence process, and even through a mesh control as a function of the geometry, which can lead to a more accurate solution.

In this work, regarding the mesh discretization, a finite element size of 3 mm was considered, where we chose to control the mesh as a function of the geometry to obtain a homogeneous mesh with a reasonable number of elements inside the dowel pins. For the representation of the mesh in three dimensions, the element SOLID 278 [16] was used with eight nodes, as indicated for the transient thermal analysis.

The nonlinearity due to the dependence on the thermal properties of the material was considered in the numerical formulation. To reach the convergence of the solution for the numerical problem, ANSYS® [16] uses the Newton–Raphson method, which determines the new equilibrium position in an incremental and iterative way.

To calculate the transient thermal response, a time-step increment of 60 s was imposed for a minimum of 1 s if there was no convergence of the solution. To optimize model convergence, an absolute tolerance of 0.9 was considered for calculating the heat flux, with the other variables being considered as standard ANSYS® parameters. At the end of the thermal analysis, the evolution of the temperatures in the model was obtained.

Due to the geometry symmetry, the numerical calculation was performed for one-half of the 3D connection subjected to fire. The results allow for the verification of the influence of the steel material in the vicinity of the wooden elements, and, at the same time, for the calculation of the charred layer produced by the wooden elements.

Figure 8 represents the meshes of the three numerical models under study. The material properties of wood and steel are represented in different colors, blue and purple, for each one. A total of nine models were simulated according to the three chosen glulam.

In the simulations, the boundary conditions are associated with convection and radiation due to exposed fire (standard fire curve ISO 834 [17], Figure 9) on two sides of the external surfaces. The numerical model has an initial temperature considered equal to 20 °C. The emissivity of the environment is a constant value and equal to 1, and the convection coefficient is equal to 25 W/m² K, according to Eurocode 3 part 1–2 [15].

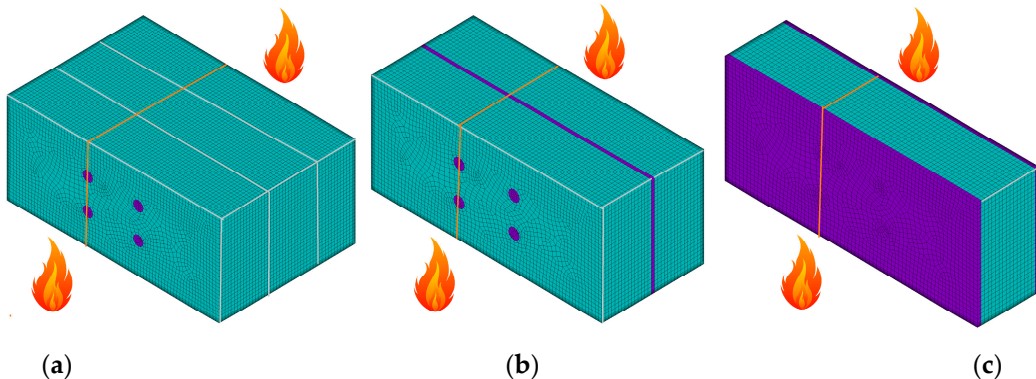

**Figure 8.** 3D Meshes, the cross-section marked as the study plane, and the fire location: (**a**) timber-to-timber, (**b**) steel-to-timber connections with internal steel plate, (**c**) steel-to-timber connections with external steel plate.

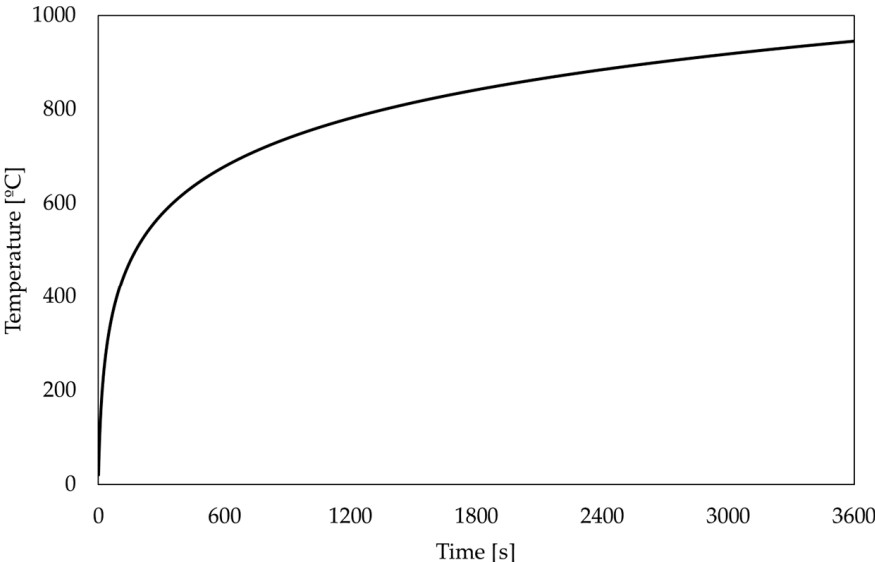

**Figure 9.** Nominal temperature–time curve ISO 834.

## 4. Results and Discussion of the Studied 3D Timber Connections

*4.1. Temperature–Time History*

Figures 10–12 represent the results of the temperature field in four specific positions within the model in the cross-section under study (K1_a, K1_b, K2_a, K2_b). These specific points are represented in these same graphs. The graphs represented have the same time scale up to 1800 s. The results are plotted across all connections in the study as a function of glulam density. In all curves, there is a similar trend for the same analysis points between the three glulam densities used. The GL20H connection appears with the highest temperatures on time, followed by the connection with GL24H and GL32H.

In timber-to-timber connection at the nodal points under analysis, after 1300 s, the temperature reaches 300 °C at point K1_a. The connection made with GL20H wood always presents the greatest temperatures over time, followed by the connection with GL24H and GL32H wood at the points studied. K1_a and K1_b represent the temperature in the vicinity of the dowel pin steel, with higher temperatures compared to positions K2_a and K2_b. Regarding K2_a, after 1500 s, the GL32H wood exceeds 300 °C. K2_b always remains at low temperatures.

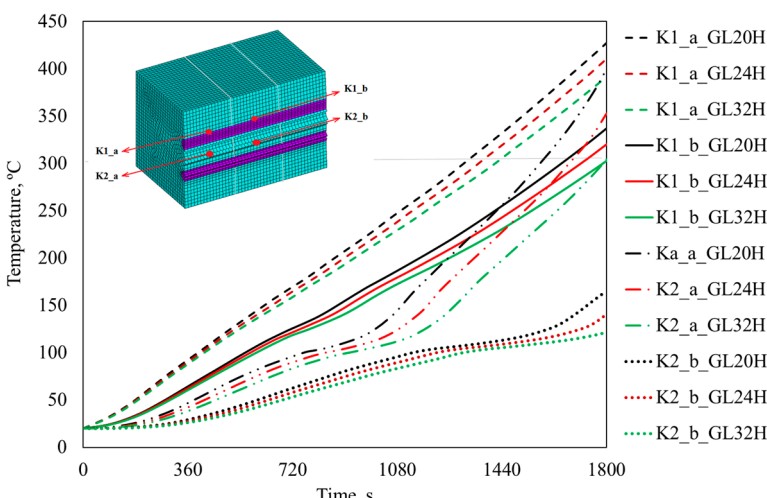

**Figure 10.** Temperature–time history in timber-to-timber connection.

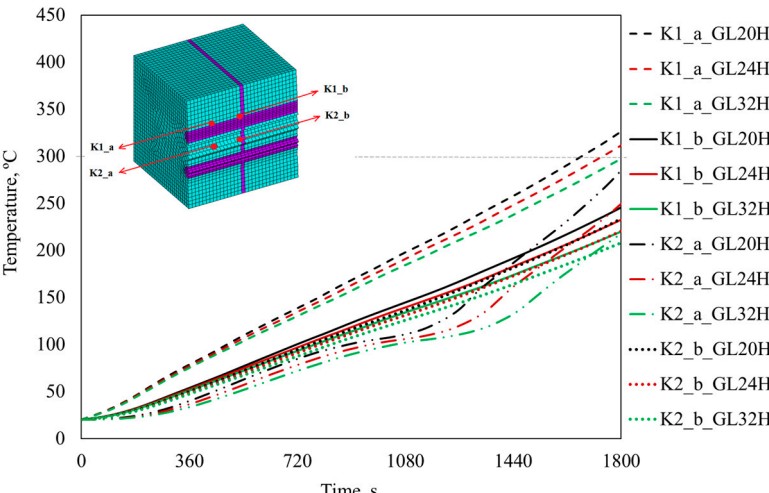

**Figure 11.** Temperature–time history in steel-to-timber connections (internal steel plate).

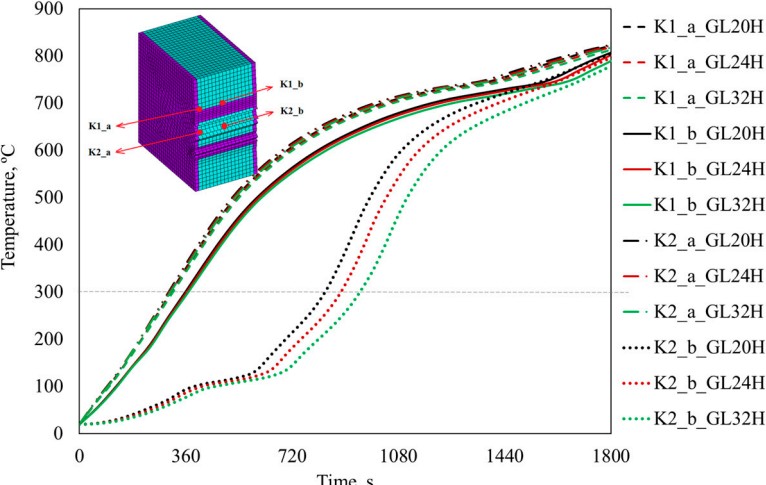

**Figure 12.** Temperature–time history in in steel-to-timber connections (external steel plate).

In steel-to-timber connections with an internal steel plate, and at the points studied, only after 1600 s does the temperature reach 300 °C at point K1_a for the GL20H wood connection. Of the points under analysis, point K1_a maintains a higher temperature

than the other points studied. These points, adjacent to the wood, receive the heat produced by the dowel pin and the internal steel plate as a heat sink, where the temperature increases slowly.

In the steel-to-timber connection with an external steel plate, all nodal points studied at the end of 1800 s of exposure to fire exceeded 300 °C, regardless of wood density. Of the points under analysis, point K2_b remained for the longest time at a temperature below 300 °C. After 300 s, the temperature reached 300 °C at points K1_a and K2_a for any type of wood connection, with a delay of 400 s for the internal points where temperature adjustment was obvious due to the glulam density effect, with a delay between the connection on the GL20H and the others. Positions K1 and K2_a close to the steel plate absorbed higher temperatures throughout the fire exposure.

Figure 13 represents the temperature in the cross-section under study for each connection in GL20H at 1800 s of fire exposure. The focus is to compare the temperature distribution within the connection where the influence of both materials is visible.

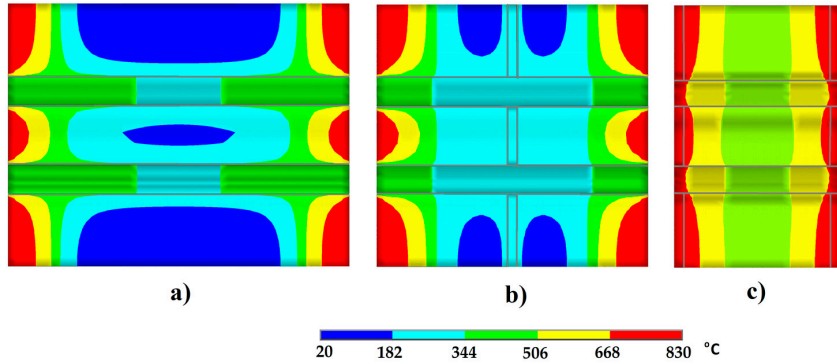

| | | |
|:---:|:---:|:---:|
| a) | b) | c) |

20       182       344       506       668       830  °C

**Figure 13.** The temperature at 1800 s of fire exposure, connections in GL20H: (**a**) timber-to-timber, (**b**) steel-to-timber connections with internal steel plate, (**c**) steel-to-timber connections with external steel plate.

In the steel-to-timber connection, the temperature reaches a lower temperature when compared to the other connections. The steel-to-timber connection reaches high temperature values because of the heating of the steel elements in front of the fire. Comparing the wooden elements between (a) and (b), adjacent to the dowel pins, the temperature is higher in (a). In (b), the internal steel plate acts as a heat sink effect that absorbs the heat from the fire through the dowel pins leading to a decrease in temperature in the wooden element in the vicinity of the dowel. These results are confirmed by the previous graphs represented in Figures 10–12, for an exposure time equal to 1800 s, and are possible to verify in Figures 14–16 for temperature calculation through the dowel pin's length.

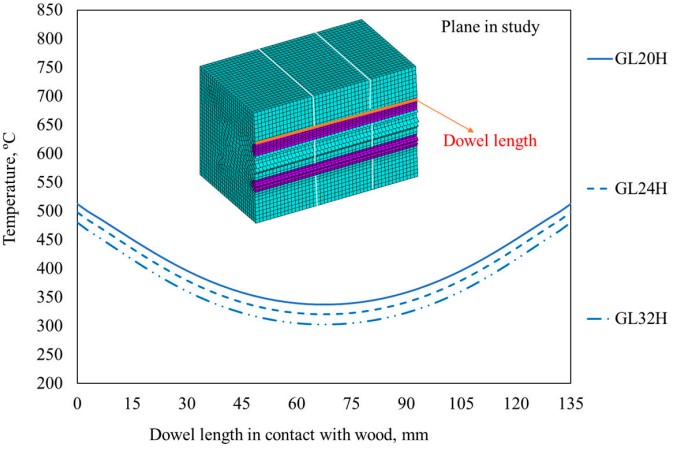

**Figure 14.** Temperature in the dowel pin length at 1800 s for timber-to-timber connection.

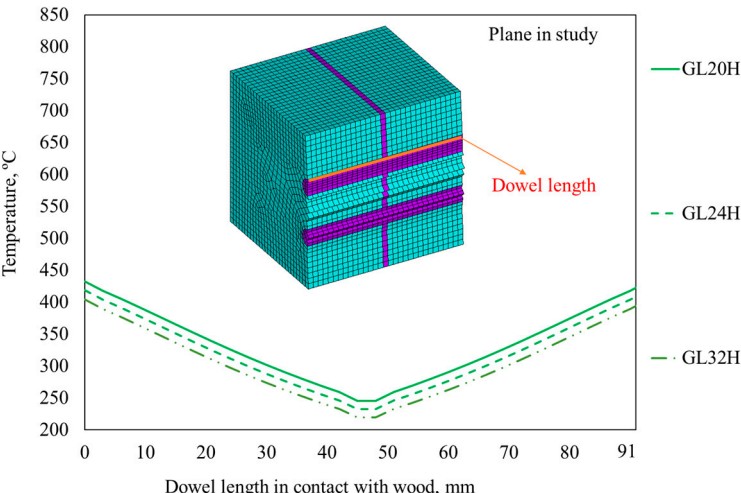

**Figure 15.** The temperature in the dowel pin length at 1800 s for steel-to-timber connections (internal steel plate).

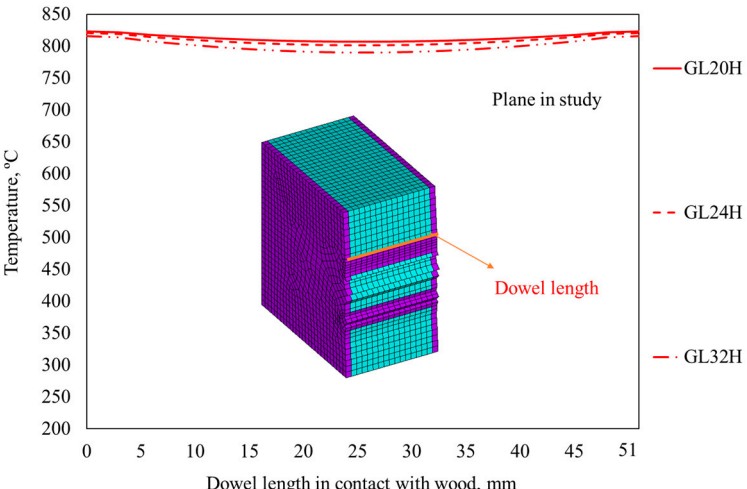

**Figure 16.** The temperature in the dowel length at 1800 s for steel-to-timber connection (external steel plate).

*4.2. Temperature Evolution throughout Dowel Pin Length*

The following results show the temperature development along the length of the dowel pin in the mean cross-sectional plane over 1800 s of fire exposure. The main objective is to find the difference between the external and internal sides in contact with the wooden component, shown in Figures 14–16.

In Figure 14, the results indicate that there is a temperature difference between 250 and 300 °C in the dowel length for the time of 1800 s, being reduced for GL32H, and greater for GL20H wood. The temperature at the edges of the dowel pin in contact with the fire is higher than in the middle because of contact with the less conductive wood material.

In Figure 15, the temperature variation was 200 °C throughout the steel dowel pin, which was also lower for GL32H wood, and higher for GL20H wood. The lowest temperature was in the middle, which is the result of the steel plate, and because of heat absorption, as was also concluded [18,19]. This connection had lower temperatures along the length of the dowel pin compared with other connections.

The results presented in Figure 16 show that there is an insignificant temperature variation throughout the dowel. In the steel-to-timber connection with external steel plates, the temperature along the dowel pin length is higher and almost constant. This type of connection had higher temperatures (around 800 °C) inside the dowel when compared to

the other connections under study, which indicates that after the 1800 s the connection was already completely charred.

### 4.3. Charred Wood Formation

Figure 17 presents the charred wood at the end of 900 s of exposure to fire in the connections studied (Figure 2), shown in gray color. Charred depth is developed when the target temperature of 300 °C is reached [14].

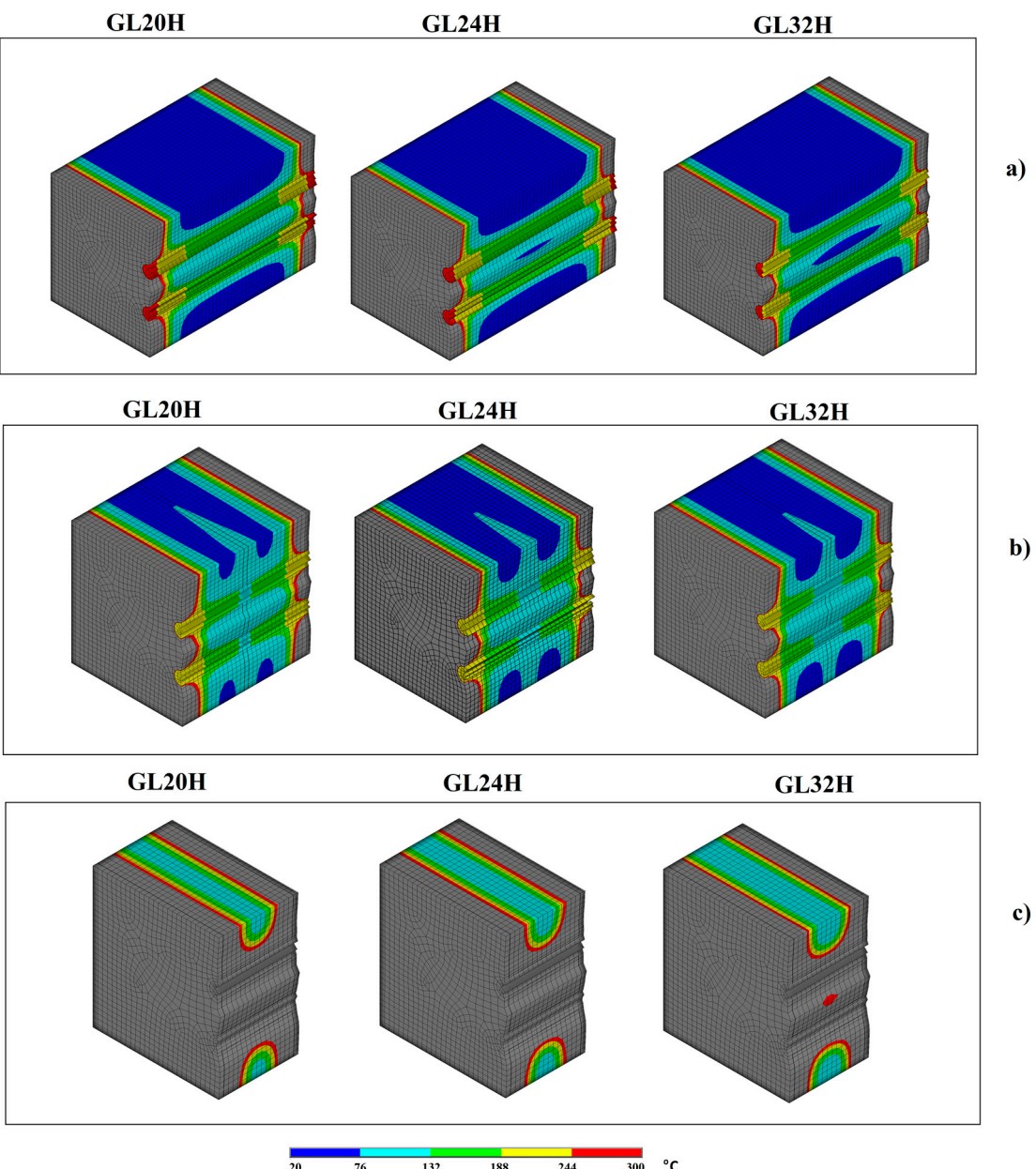

**Figure 17.** The temperature at 900 s of fire exposure, function of the density material: (**a**) timber-to-timber, (**b**) steel-to-timber connections with internal steel plate, (**c**) steel-to-timber connections with external steel plate.

The remaining area of wood away from the dowels has lower temperatures than the residual area next to the dowels. Additionally, GL20H timber connections experience higher internal temperatures than GL32H connections. It is promising to observe that although the steel provides greater flow within the connection, the wooden elements provide some protection at the beginning of the process. This means that both elements contribute to the

evolution of the charred area in the connection. With 900 s of exposure to fire, the steel plate as a core member has lower temperatures when compared to the connection with external steel plates. As for the charred area, it slowly decreases with exposure to fire because the evolution on the surface of the wood prevents the fire from advancing, so the remaining material does not change [13].

Based on three chosen instants in time, the charred layer was measured in each connection, based on the size of each element (3 mm). Different measured points were considered in the middle of the glulam element from the top to the inside. Then, the average charring rate of each one was calculated, varying the glulam density, as seen in Table 3. After an analysis of the values obtained and a comparison with that specified in Eurocode 5 part 1–2 [14], equal to 0.65 mm/min, some assumptions can be considered. The charring rate is not constant and depends on the type of glulam.

**Table 3.** The average charring rate for each type of connection, mm/min.

| Connection Type | GL20H | GL24H | GL32H |
|---|---|---|---|
| Timber-to-timber | 0.80 | 0.75 | 0.69 |
| Steel-to-timber (internal steel plate) | 0.85 | 0.78 | 0.73 |
| Steel-to-timber (external steel plate) | 0.86 | 0.80 | 0.74 |

The higher density glulam has a lower charring rate. The steel elements in the connections promote the evolution of heat inside the wooden element, where the calculated charring increases.

## 5. Conclusions

As calculated, and according to the simple Eurocode equations for ambient temperature (Table 2), the steel-to-timber connection with an external steel plate with high density material has the highest load-carrying capacity, and the timber-to-timber connection with lower density material has a lower strength capacity.

However, in fire conditions, and according to the numerical simulations, the following can be concluded: in timber-to-timber connections in GL20H wood, temperatures have higher values compared to GL32H wood connections; inside the wood element, the temperature remains lower; the steel dowel pins affect the heat produced inside the timber connection and remain at high temperatures. The steel-to-timber connection with the external steel plate subjected to fire shows the temperature that is highly developed through the steel elements, which indicates that after the 1800 s the connection was already completely charred and lost all its strength.

As a proposal for future work, it would be interesting to create new 3D numerical models varying other factors, such as the diameter of the dowel pins, the thickness of the steel plates, the number, and the position of all the connectors. In addition, faced with the need to find alternatives and with the application of these elements in construction needing to comply with safety requirements, namely in situations of accidental action, it is important to continue the study of models of protected connections. Additionally, the presented methodology should be validated with experimental tests, and we should continue to study and develop the thermomechanical analysis to compare different loaded elements.

**Author Contributions:** Conceptualization, E.M.M.F.; methodology, E.M.M.F.; writing—review and editing, E.M.M.F.; validation, C.G.; investigation, C.G.; writing—original draft, C.G. All authors have read and agreed to the published version of the manuscript.

**Funding:** This research received no external funding.

**Informed Consent Statement:** Not applicable.

**Data Availability Statement:** Not applicable.

**Conflicts of Interest:** The authors declare no conflict of interest.

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
