# Peer review of "FEM Analysis of 3D Timber Connections Subjected to Fire: The Effect of Using Different Densities of Wood Combined with Steel"

_fire, doi:10.3390/fire6050193_

Round 1
Reviewer 1 Report
The results of the analysis can be translated into more conclusions. In particular, I propose to present the change in the loadbearing capacity of each type of connection over time. This would allow for additional assessment of the rate of decrease of loadbearing capacity.
Captions under Figs. 3 to 7 should be corrected.
Author Response
Comments and Suggestions for Authors
The results of the analysis can be translated into more conclusions. In particular, I propose to present the change in the loadbearing capacity of each type of connection over time. This would allow for additional assessment of the rate of decrease of loadbearing capacity.
Captions under Figs. 3 to 7 should be corrected.
ANSWER – The authors would like to acknowledge the Reviewer for their work on reading and suggesting improvements to the manuscript. We have addressed all the comments and the changes needed. Changes to the article were identified in green.
Reviewer 2 Report
In this paper, the thermodynamic properties of steel and wood structures with three different densities of wood are analyzed by finite element method, which should be used for the safety design of typical steel-wood joints under fire. However, there are several questions not mentioned or clearly clarified by the authors, and therefore the manuscript is not recommended for publication.
1) The notes for Figures 3 to 7 are identical, and these five figures have been used in the previously published article ‘Thermomechanical Analysis of Steel-to-Timber Connections in Tensile and Exposed to Fire: The Wood Density Effect’.
2) It necessary to supplement the thermal conductivity of the three different woods used in the experiment.
3) Is there a contradiction between the highest temperature at position K1_b in Figure 11 and the lowest temperature at position K1_b in Figure 14?
4) In the abstract of the text, the authors state ‘Steel-to-timber connections with an internal steel plate with high glulam density, have higher fire resistance due to the obtained lower temperatures’, However, at the conclusion the author states ‘the steel-to-timber connection with an external steel plate with high density material has the highest load carrying capacity’, so what is the final conclusion obtained by simulation?
5) There is an error in the spelling of the ‘°C’ symbol in the text; there is an error in the punctuation on page 8, line 206; there is a spelling error in the figure note of Figure 12 on page 10, line 234; there is a spelling error in GL32H in Figure 16 on page 12, please check and correct it carefully.
Minor editing of English language required
Author Response
About the Reviewer’s comments, we are answering their concerns in the following lines. The authors would like to acknowledge the Reviewer for their work on reading and suggesting improvements to the manuscript. We have addressed all the comments and the changes needed. Changes to the article were identified in green.
1) The notes for Figures 3 to 7 are identical, and these five figures have been used in the previously published article ‘Thermomechanical Analysis of Steel-to-Timber Connections in Tensile and Exposed to Fire: The Wood Density Effect’.
ANSWER – The authors thank the reviewer. The corrections were introduced. The reference [4] was introduced before the five Figures.
2) It necessary to supplement the thermal conductivity of the three different woods used in the experiment.
ANSWER – The thermal conductivity of the three different kinds of wood is in accordance with Eurocode 5, part 1-2 (Annex B).
3) Is there a contradiction between the highest temperature at position K1_b in Figure 11 and the lowest temperature at position K1_b in Figure 14?
ANSWER – Point K1_b in Figure 11 is in the wood in the vicinity of the steel plate. There was a change between K1_a and K1_b, due to a mistake, now been corrected. In Figure 15, the lowest temperature is in the middle of the connection on the dowel pin surrounded by the steel plate.
4) In the abstract of the text, the authors state ‘Steel-to-timber connections with an internal steel plate with high glulam density, have higher fire resistance due to the obtained lower temperatures’, However, at the conclusion the author states ‘the steel-to-timber connection with an external steel plate with high density material has the highest load carrying capacity’, so what is the final conclusion obtained by simulation?
ANSWER – The authors reorganized the writing of the conclusions.
5) There is an error in the spelling of the ‘°C’ symbol in the text; there is an error in the punctuation on page 8, line 206; there is a spelling error in the figure note of Figure 12 on page 10, line 234; there is a spelling error in GL32H in Figure 16 on page 12, please check and correct it carefully.
ANSWER – The authors thank the reviewer. The corrections were introduced.
Reviewer 3 Report
Dear Authors,
please find my comments and suggestions in the attached pdf. Overall, I found the manuscript interesting, but results could be presented more clearer and in some more detail and supported or compared with other references. I also think that the current title of the Technical Note is not representative of the described study and I would strongly suggest rewriting it. I find the work important for the field and that is why I think that additional work on the manuscript could add value to the manuscript.
All best.

Dear Authors,
I am not an English native speaker, but I still think that the quality of the English language and readability could and must be improved. In the text, I suggested some changes, where I could find better sentence structure, but the manuscript could benefit if some professional would improve the text.
Author Response
Dear Authors,
please find my comments and suggestions in the attached pdf. Overall, I found the manuscript interesting, but results could be presented more clearer and in some more detail and supported or compared with other references. I also think that the current title of the Technical Note is not representative of the described study and I would strongly suggest rewriting it. I find the work important for the field and that is why I think that additional work on the manuscript could add value to the manuscript.
All best.
ANSWER – About the Reviewer’s comments, we are answering their concerns in the following lines. The authors would like to acknowledge the Reviewer for their work on reading and suggesting improvements to the manuscript. We have addressed all the comments and the changes needed. Changes to the article were identified in green.
Reviewer 4 Report
The technical note “3D Timber Connections Subjected to Fire, Using Different Densities Combined with Steel Material” describes some F.E.M. simulations (Finite Element Method), with the aim to understand and predict the behaviour of different kind of timber connection. The theoretical research has a good setting up. The paper is clear in its objectives and approach. Results seem enough interesting.
However, the reviewer found some unclear and/or potentially wrong points that need some careful clarification or correction.
The results showed in figures 10 and 11 present some anomalies. Specifically, the points K2_a in the two experimental sets, which have potentially the same characteristics in relation to the fire exposition, really show two different behaviour: in fig. 10, after 1800 seconds the values of the point K2_a are in the range between 300 and 400 °C, while in fig. 11 the same point K2_a is in the range 200-300 °C (approximately). These different behaviours, for the same kind and time of exposure, for the reader are incomprehensible.
Likewise, the figures 13 and 14, where the temperature of the two dowel pins are showed. It seems quite strange that after 1800 seconds a difference of around 80 °C could be possible, since the simulated external conditions are virtually the same.
In any case:
1) either there is a mistake and therefore the experimental results and the paper must be corrected,
2) or the reviewer is wrong (e.g. is there perhaps a different extraction of heat from the environment by one joint compared to the other to explain the differences? but in the conditions of a building fire the heat subtraction by objects is absolutely insignificant…), then the meaning of the FEM test must be better explained and the paper corrected.
Furthermore, the title and the abstract do not explain in a sufficient way the content of the paper, specifically the reader cannot assume that the article deals only with FEM simulations. The reviewer, reading firstly the title and the summary, thought that they were experimental tests carried out in the laboratory, and not just FEM simulations.
Author Response
The technical note “3D Timber Connections Subjected to Fire, Using Different Densities Combined with Steel Material” describes some F.E.M. simulations (Finite Element Method), with the aim to understand and predict the behaviour of different kind of timber connection. The theoretical research has a good setting up. The paper is clear in its objectives and approach. Results seem enough interesting.
However, the reviewer found some unclear and/or potentially wrong points that need some careful clarification or correction.
The results showed in figures 10 and 11 present some anomalies. Specifically, the points K2_a in the two experimental sets, which have potentially the same characteristics in relation to the fire exposition, really show two different behaviour: in fig. 10, after 1800 seconds the values of the point K2_a are in the range between 300 and 400 °C, while in fig. 11 the same point K2_a is in the range 200-300 °C (approximately). These different behaviours, for the same kind and time of exposure, for the reader are incomprehensible.
Likewise, the figures 13 and 14, where the temperature of the two dowel pins are showed. It seems quite strange that after 1800 seconds a difference of around 80 °C could be possible, since the simulated external conditions are virtually the same.
In any case:
1)either there is a mistake and therefore the experimental results and the paper must be corrected,
2)or the reviewer is wrong (e.g. is there perhaps a different extraction of heat from the environment by one joint compared to the other to explain the differences? but in the conditions of a building fire the heat subtraction by objects is absolutely insignificant…), then the meaning of the FEM test must be better explained and the paper corrected.
Furthermore, the title and the abstract do not explain in a sufficient way the content of the paper, specifically the reader cannot assume that the article deals only with FEM simulations. The reviewer, reading firstly the title and the summary, thought that they were experimental tests carried out in the laboratory, and not just FEM simulations.
ANSWER –
About the Reviewer’s comments, we are answering their concerns in the following lines. The authors would like to acknowledge the Reviewer for their work on reading and suggesting improvements to the manuscript. We have addressed all the comments and the changes needed. Changes to the article were identified in green.
The manuscript was rewritten. The title of the manuscript was changed. The abstract is now more concise, with more explanations introduced. Regarding the results in Figures 10-12 and 14-16, more explanations were introduced in the manuscript to justify the level of the obtained temperatures and the comparison between all connections. A new Figure 13 was introduced to justify the temperature within the cross-section to see the effect in the wooden elements, due to the internal steel plate acting as a heat sink effect that absorbs the heat from the fire through the dowel pins leading to a decrease in temperature in the wooden element in the vicinity of the dowel. These results are confirmed by the previous graphs for an exposure time equal to 1800 s and possible to verify temperature calculation through the dowel pin length.
Reviewer 5 Report
- Correct the description of "Figure 2" to "Figure 7".
- It was observed that the aim of the article is to present an estimated thermal behavior of the connections by means of preliminary results and, for a while, without any validation based on comparison with other numerical or experimental research. Therefore, the validation of the presented method, as well as thermomechanical analysis could be important proposal of future work.
Author Response
- Correct the description of "Figure 2" to "Figure 7".
- It was observed that the aim of the article is to present an estimated thermal behavior of the connections by means of preliminary results and, for a while, without any validation based on comparison with other numerical or experimental research. Therefore, the validation of the presented method, as well as thermomechanical analysis could be important proposal of future work.
ANSWER –
The authors would like to acknowledge the Reviewer for their work on reading and suggesting improvements to the manuscript. We have addressed all the comments and the changes needed. Changes to the article were identified in green.
All the corrections were introduced in the Figures. In the conclusions, more future work was introduced.
Round 2
Reviewer 2 Report
accept
Author Response
The authors would like to thank again the acknowledge to the Reviewer for their continuing work on reading our manuscript.
Reviewer 3 Report
Dear Authors,
thank you for revising the manuscript. In my opinion, the quality now improved.
At this time I noticed that in line 217 is still the wrong symbol for the degree and in some cases, the number and units for degrees are together, and in some cases separated by space. I would recommend uniforming the style.
The quality of the Figures is much worse now than in the previous manuscript and must be replaced.
Best.
Author Response
The authors would like to thank again the acknowledge to the Reviewer for their continuing work on reading our manuscript. All the corrections were introduced. About the Figures, in the Word manuscript, they appear with high quality. The previous PDF file was automatically generated by the system. Now, I will try to submit both format files.
Reviewer 4 Report
Text editing and corrections were appreciated. Very good.
Author Response

(The authors gave the same response as above.)
